# Percolation Conduction of Carbon Nanocomposites

**DOI:** 10.3390/ijms21207634

**Published:** 2020-10-15

**Authors:** Grigorii S. Bocharov, Alexander V. Eletskii

**Affiliations:** 1Institute of Thermal and Nuclear Power Engineering, National Research University MPEI, 111250 Moscow, Russia; eletskii@mail.ru; 2Joint Institute of High Temperatures RAS, 125412 Moscow, Russia

**Keywords:** polymer nanocomposites, carbon nanotubes, graphene oxide, percolation conduction

## Abstract

Carbon nanocomposites present a new class of nanomaterials in which conducting carbon nanoparticles are a small additive to a non-conducting matrix. A typical example of such composites is a polymer matrix doped with carbon nanotubes (CNT). Due to a high aspect ratio of CNTs, inserting rather low quantity of nanotubes (on the level of 0.01%) results in the percolation transition, which causes the enhancement in the conductivity of the material by 10–12 orders of magnitude. Another type of nanocarbon composite is a film produced as a result of reduction of graphene oxide (GO). Such a film is consisted of GO fragments whose conductivity is determined by the degree of reduction. A distinctive peculiarity of both types of nanocomposites relates to the dependence of the conductivity of those materials on the applied voltage. Such a behavior is caused by a non-ideal contact between neighboring carbon nanoparticles incorporated into the composite. The resistance of such a contact depends sharply on the electrical field strength and therefore on the distance between neighboring nanoparticles. Experiments demonstrating non-linear, non-Ohmic behavior of both above-mentioned types of carbon nanocomposites are considered in the present article. There has been a model description presented of such a behavior based on the quasi-classical approach to the problem of electron tunneling through the barrier formed by the electric field. The calculation results correspond qualitatively to the available experimental data.

## 1. Introduction

Carbon nanocomposites present a new class of nanomaterials in which conducting carbon nanoparticles are a small additive to a non-conducting matrix. Nanocomposites on the basis of polymers doped with carbon nanotubes (CNT) present a typical example of such systems (see, for example Ref. [1,2,3,4] and the literature cited there). In such composites, at exceeding some critical value of CNT concentration, the percolation transition occurs which accompanies with the enhancement of the conductivity of the material by 10–12 orders of magnitude. Due to a high aspect ratio of CNTs (ratio of the length to the diameter ~10^3^–10^4^) such a transition proceeds at very low concentrations of nanotubes (on the level of 0.01%). Electrical properties of polymer composites doped with CNTs were studied and described in a great quantity of publications (see, for example the reviews [1,2,4]). These review articles contain experimental data on the conductivity of a large number of polymer composites with various types of polymer matrices. However, these data relate to certain values of the applied voltage, while the dependence of the conductivity on the applied voltage noted in some experiments publications has not taken into account. In the present article, the character of the dependence of the conductivity of polymer nanocomposites doped with CNTs on the applied voltage is analyzed. The results of a simple model description of such a dependence are compared with available experimental data.

Another type of carbon nanocomposite is represented by a film consisted of fragments of partially reduced graphene oxide (GO). Such a film is formed particularly as a result of thermal reduction of GO [5,6,7]. In such composites, the percolation transition proceeds not as a result of the enhancement of the concentration of conducting nanoparticles, but due to the increase in conductivity of such particles as a result of the reduction process. Such a behavior can be considered as a new type of percolation transition. The distinctive peculiarity of both types of nanocomposites is in the dependence of the conductivity of the material on the applied voltage [8,9,10,11,12,13,14,15,16,17,18,19,20,21]. Such a behavior is caused by a non-ideal character of the contact between the neighboring carbon nanoparticles involved in the composite. The value of the contact resistance is very sensitive to the electrical field, which results in the dependence of the conductivity of the material on the applied voltage.

The present article contains results of theoretical and experimental studies of non-Ohmic behavior of the electric resistivity of the above-noted materials. Specifically, electric characteristics of polymer composites doped with CNTs have been analyzed. The state of the system in a vicinity of the percolation threshold has been considered in detail, when the charge transport is performed by one or few percolation paths. The current-voltage characteristics of a percolation chain consisting of CNTs are calculated assuming that the contact resistance between the neighboring nanotubes is much higher than the intrinsic resistance of CNTs. The dependence of the contact resistance on the applied electrical field or inter-tube distance is described on the basis of the known quasi-classical approach. In accordance with this approach, the electron transport proceeds as a result of tunneling through the barrier formed by the electric field. The distance between neighboring contacting nanotubes is supposed to obey the random (normal) distribution. The calculations performed result in a non-linear, non-Ohmic form of current-voltage characteristics of the material. Experiments demonstrating non-Ohmic behavior of nanocomposites doped with CNTs have been analyzed. The current-voltage characteristics measured in these experiments are compared with the results of calculations performed.

The procedure of thermal reduction of graphene oxide (GO) has been described in detail. This procedure results in a set of samples differing in the conductivity, which is determined by the thermal processing temperature. The measured dependences of the conductivity and density of the partially reduced GO samples on the thermal processing temperature have been discussed. The measurements indicate that the percolation transition occurs at the annealing temperature of about 170 °C. This transition proceeds when the concentration of conducting graphene flakes reaches a value at which one or few conducting paths form. The number of such paths enhances as the annealing temperature increases, so that at the annealing temperature of about 800 °C, graphene oxide reduces practically completely, and the conductivity of the material approaches the reference value for graphite. The non-linear electric behavior of GO samples treated at various temperatures has been studied. The measurements performed indicate a notable dependence of the degree of non-linearity on the annealing temperature. Such a behavior is explained in terms of different inter-relation between the characteristic values of the resistance of partially reduced GO flakes and contacts between them in dependence on the annealing temperature. At relatively low annealing temperatures (lower ~300 °C) the contact resistance is much less than the characteristic resistance of GO flakes, so that the resistivity of a sample is determined by the resistance of GO fragments, which does not practically depend on the applied voltage. Enhancement of the annealing temperature is accompanied by the increase in the conductance of GO fragments, so that the conductivity of GO samples is determined mainly by the contact conductance, which is overly sensitive to the value of the applied voltage. A model description of non-linear electric characteristics of partially reduced GO samples corresponds qualitatively to measurement results.

## 2. Percolation Behavior of Polymer-Based Composites Doped with CNTs

### 2.1. Model of Charge Transport

Theoretical description of electric characteristics of such composites is usually based on the Monte-Carlo method. In accordance with this approach, the percolation chain is formed as a result of a random distribution of nanotubes over polymer matrix bulk. Therewith the position of the percolation threshold is roughly inversely proportional to the aspect ratio of nanotubes comprising the percolation chain [1,2,3]. Such a dependence is derived from the percolation theory, and the comparison of results of model calculations with the relevant experimental data usually results in a good coincidence for the percolation threshold values of such materials. However, the absolute values of the conductivity in a vicinity of the percolation threshold, calculated on the basis of the Monte-Carlo method, as a rule, exceed those measured by various authors by 2–4 orders of magnitude [1,2]. Analysis performed in Refs. [19,20,21] has shown that such a disagreement is caused by the existence of non-ideal contacts between the nanotubes forming the percolation chain. The contact resistance usually exceeds the typical resistance of nanotubes by several orders of magnitude. Therewith the conductivity of a polymer-based composite doped with CNTs is determined practically fully by the contact resistance between nanotubes. The charge transport between the nanotubes in contact with each other in a non-ideal manner proceeds by electron tunneling through the potential barrier, forming as a result of action of the electric field. The tunneling probability depends sharply (exponential) on both the distance between nanotubes and the contact voltage. Besides of that inside the contact can be found one or few polymer molecules hindering the charge transport process [1,2].

Applying external electric field changes the form of the potential barrier separating two conducting nanoparticles so that the probability of electron tunneling under the barrier increases sharply as the electrical field enhances. This causes a dependence of the conductivity of polymer-based materials doped with CNTs on the applied voltage and non-Ohmic behavior of such materials. One should note that polymer-based composites doped with CNTs have a complicated structure. The contact resistance between neighboring nanotubes can depend not only on the distance between them but also on the contact angle, on the type of polymer molecules found in a vicinity of the contact and on the orientation of such molecules. The uncertainty of those factors hinders and practically excludes comprehensive quantitative determination of the dependence of the contact resistance on the applied voltage. The usage of a simple model approach, in which validity can be estimated on the basis of comparison with experiment, seems to be natural in such a situation [19,21]. In accordance with this approach [19,21], the conduction electrons are found in a rectangular potential well in which the depth is determined by the work function *φ*_e_. The value of this parameter is close to 5 eV [22], which exceeds the characteristic value of the kinetic energy of conduction electrons (*E*_e_~300 K). Therefore, electrons perform the tunnel transition from one to another nanotube through the potential barrier of *φ*_e_—*E*_e_ in height. Application of the electric field *F* changes the shape of this barrier (Figure 1) in accordance with the known result of the quasi-classical approximation [23]. The tunneling probability for electron with the energy *E*_e_ is expressed through the parameters of the barrier and electrical field as follows:(1)$$W\approx exp\left[-\frac{2}{\hbar }\int_{0}^{d}\sqrt{2m\left(U\left(x\right)-E_{e}\right)}dx\right]$$
where $U\left(x\right)=\varphi_{e}-eFx$; *m* and *e* are the electron charge and mass, correspondingly; *d* is the barrier width, which is determined by the distance between the nanotubes. Integration (1) results in the following expression:(2)$$W\approx exp\left[-\frac{4\sqrt{2m}}{3\hbar eF}{\left(\varphi_{e}-E_{e}\right)}^{3/2}\left(1-{\left(1-\frac{eF}{\varphi_{e}-E_{e}}d\right)}^{3/2}\right)\right]$$

The contact resistance *R*_c_ is inversely proportional to the tunneling probability and can be represented in the following form:(3)$$R_{c}=R_{0}exp\left[\frac{4\sqrt{2m}}{3\hbar eF}{\left(\varphi_{e}-E_{e}\right)}^{3/2}\left(1-{\left(1-\frac{eF}{\varphi_{e}-E_{e}}d\right)}^{3/2}\right)\right]$$
where *R*_0_ is the contact resistance at a zero distance between the nanotubes. The value of this parameter depends, in a complicated manner, on the chirality and other geometrical features of the contacting nanotubes [1,24]. For definiteness, this value can be supposed to be equal ballistic resistance $R_{0}=\hbar /2e^{2}=12.94\;k\Omega$. At such a supposition the spread in values of the contact resistance is determined by the spread in the inter-tube distances. The exponential dependences (2) and (3) are very sensitive to the value of this distance.

At a moderate excess of the percolation threshold there exists one or few conducting paths consisting of nanotubes closely placed to each other. In this case, the resistance of the nanocomposite *R* is the sum of resistances *R*_i_ of nanotubes forming the percolation path, and the relevant contact resistances *R*_ci_:(4)$$R=\overset{n}{\underset{i}{\sum }}\left(R_{i}+R_{ci}\right)$$

The resistance of nanotubes is usually much less than the contact resistance, *R*_i_ ≪ *R*_ci_, so that the resistance of nanotubes in Equation (4) can be neglected. If this inequality is violated for some contact, two contacting nanotubes should be considered as a single one.

Taking into account the above-given inequality and Equation (3) one obtains from Equation (4):(5)$$R=R_{0}\overset{n}{\underset{i=1}{\sum }}exp\left[\frac{4\sqrt{2m}}{3\hbar eF_{i}}{\left(\varphi_{e}-E_{e}\right)}^{3/2}\left(1-{\left(1-\frac{eF_{i}}{\varphi_{e}-E_{e}}d_{i}\right)}^{3/2}\right)\right]$$
where *d*_i_ is the inter-tube distance for *i*-th contact, $F_{i}=U_{i}/R_{ci}$ is the electric field strength on *i*-th contact, and *U*_i_ is the potential drop on *i*-th contact.

The current *I* through a percolation channel is expressed through the contact resistance by the obvious relation
(6)$$U_{i}=IR_{ci}$$

Under conditions of percolation the current-voltage characteristic (CVC) of a nanocomposite is expressed through the percolation channel current with taking into account the distribution of inter-tube distances *d*_i_,
(7)$$U=\overset{n}{\underset{i=1}{\sum }}U_{i}.$$

In accordance with Equation (5), the resistance of a percolation chain depends notably on the applied voltage. Therewith the electric behavior of polymer-based composites doped with CNTs has a non-Ohmic character.

The statistical spread of the inter-tube distances *d* is naturally to be modelled by the normal distribution
(8)$$P\left(d\right)=\frac{1}{\sqrt{2\pi }\sigma }exp\left[-\frac{{\left(d-\mu \right)}^{2}}{2\sigma^{2}}\right]$$
where *P*(*d*) is the probability density of the specific distance normalized by unity, μ is the average value of this parameter, and *σ* is its standard deviation. Taking into account Equation (8), Equation (7) has the following form:(9)$$U=\overset{n}{\underset{i=1}{\sum }}IR\left(d_{i}\right)P\left(d_{i}\right)$$

At calculation of Equation (9) the distances *d*_1_, *d*_2_, …, *d*_n_ were chosen randomly by means of the random number generator with taking account the distribution (8). The calculations were performed for different values *μ* and *σ*, as well as for different numbers of nanotubes *n* forming the percolation channel. The resistance was calculated several times for each pair of values *μ* and *σ* using various sets of distances *d*_1_, *d*_2_, …, *d*_n_. Figure 2 presents the dependences of the resistance of percolation chains of different length calculated for various values *μ* and *σ*. Non-linear (non-Ohmic) behavior of the nanocomposite conduction manifests itself in a notable decrease of the chain resistance as applied voltage increases. One should note that the sensitivity of the percolation chain conductivity to the applied voltage as the dispersion of the distribution of the inter-tube distance *σ* rises. At low values *σ* (*σ* ≈ 0.01 *μ*) this dependence shows itself at much higher values of the applied voltage. Note also that this effect is much more notable with a greater number of nanotubes comprising the percolation channel.

The above-stated non-Ohmic resistance of nanocomposites doped with CNTs causes non-linear CVC of such materials. Figure 3 presents the results of model calculations of this characteristic performed for different values of the average inter-tube distance *μ,* various dispersion *σ,* and various number of nanotubes *n* forming the percolation channel. As is seen, the degree of non-linearity of CVC increases as the average inter-tube distance μ enhances and the dispersion of this distance *σ* decreases.

### 2.2. Non-Ohmic Behavior of Polymer-Based Composited Doped with CNTs

The phenomenon of non-linear conduction of polymer-based nanocomposites doped with CNTs has been observed in Refs. [8,9,10,11,12,13,14,15,16,17,18]. The authors [15] utilized in their experiments a nanocomposite on the basis of polymer SU8 (a modification of epoxy resin) with admixture of multi-layer CNTs synthesized by CVD method at a temperature 640 °C with the usage of Fe-Co particles as a catalyst. The CNTs were of *d* ≈ 13.3 nm in average diameter and *L* ≈ 10 μm in length. SU8 resin doped with CNTs in the presence of a surface-active substance was subjected to ultrasonic treatment to provide a uniform distribution of CNTs over the volume. On the last stage of the nanocomposite preparation a photo initiator was inserted into the resin, which enhanced the polymerization rate. The suspension obtained was deposited onto a glass plate by means of a surgery scalpel.

Three types of samples of 2 × 1 cm^2^ in size have been prepared for further investigation. The samples of the first type (N-P) presenting non-polymerized matrix were fabricated by drying the film at a temperature of 95 °C. The samples of the second type (P) were treated at a temperature of 150 °C to provide the polymerization of SU8 resin. The samples of the third type (P-B) were obtained as a result of annealing at a temperature of 200 °C. The CNTs content in the composite samples was varied between 0.1% and 5% (by weight). The total number of samples with various concentration of CNTs accounted 19.

Figure 4 shows the results of measurements of the dependence of the conductivity of three types CNT-SU8 nanocomposite on CNT concentration [15]. The shape of these dependences is typical for the phenomenon of percolation conduction. In the case of non-polymerized resin (N-P) the percolation threshold was observed at a CNT concentration of ~10^−3^, which corresponds roughly to the inversed value of the aspect ratio of nanotubes *d/L* [1]. The conductivity of polymerized samples Р and Р-В in a vicinity of the threshold exceeds considerably that for the N-P samples. The position of the percolation threshold determined by the authors [25] who studied the percolation conduction of epoxy/multi-walled carbon nanotube (MWCNTs) composites is of the same order which indicates a similar value of the aspect ratio of CNTs used in both works.

Authors [14] have also observed non-Ohmic behavior of polymer-based nanocomposites doped with CNTs. They used as a polymer matrix high pressure polyethylene “HDPE0390” with the average molecular of 60,000. Multi-layer CNTs were synthesized by CVD method. The composites CNT-HDPE with various content of CNTs were prepared as a result of mixing the melt in an extruder. The rotation frequency of the extruder was 50 rpm, the mixing duration was 10 min. Before the electrical measurements the samples were subjected to hot pressing at a temperature of 250 °C, which resulted in formation of a thick film. Measurements have shown that the percolation threshold for the conduction of the samples was about 2.5 weight%. The current-voltage characteristics (CVC) of the material were measured for samples containing 5 and 7% CNTs in the single pulse regime to avoid effects related to the Joule heating. Figure 5 presents the results of measurements. As is seen, CVC have a clearly defined non-linear character in the range of enhanced voltages, which manifests itself in an increase of the conductivity at an enhancement of the applied voltage. The authors [14] invoke the theory of percolation networks conduction with inter-particle distances changing as a result of fluctuations [25] as the physical mechanism causing the voltage dependence the conductivity of polymer-based composites doped with CNTs.

Non-Ohmic behavior of a composite on the basis of epoxy resin doped with CNTs was studied in detail by the authors [9]. The authors separate three regions on the current-voltage characteristics of the composite. In the region of relatively low voltages, the characteristics have a linear character. At higher voltages, a sharp rise of conductivity with the increase of the applied voltage is observed. A further enhancement in the voltage is accompanied by a notable decrease in the degree of the dependence of the conductivity on the applied voltage. The authors [9] consider electron tunneling through an epoxy resin layer as well as Joule heating of the contact as physical mechanisms determining the non-Ohmic behavior of the composite. Figure 6 presents typical current-voltage characteristics of samples with different CNT content.

CNTs have a good conductivity and high aspect ratio, therefore their usage as dopant to polymer-based nanocomposites permits one to get conductive nanocomposites even at a low CNT content (at a level of 0.01%). However, such composites have rather limited conductivity due to non-ideal contact between neighboring nanotubes, which decreases their practical value. An interesting approach to the solution of this problem has been demonstrated by the authors of [17], who used as a dopant to polymers on the basis polycarbonate (РС) multi-walled carbon nanotubes whose surface is covered by a layer of conducting polymer (poly(3-4ethylenedioxythiophene)poly(styrenesulfonate)) (EPP). Therewith the non-conducting space between neighboring nanotubes is filled with conducting polymer material. To obtain the coverage, the nanotubes were purposely immersed into 5% solution of EPP in ethylene glycol. At a mass ratio ЕРР/CNTs 1.3:1 the thickness of coverage was about 10 nm.

Dependences of the conductivity on the electric field were measured for polycarbonate doped with CNTs. Rectangular samples of 60 × 10 × 0.3 mm^3^ in size contained CNTs with (E-CNT/PC) and without (CNT/PC) coverage have been prepared. The mass content of CNTs accounted for 1, 1.5, and 2%. The end surfaces of samples were covered with silver paint for performing the electric measurements. The results of such measurements obtained for samples with various CNTs content are presented in Figure 7. As is seen, the usage of CNTs with conducting coverage results in a considerable (up to 5 orders of magnitude) enhancement of the conductivity of composites. One should note that for the samples with conducting coverage, the conductivity does not practically depend on the electric field. This demonstrates that the usage of a conducting coverage provides practically ideal contact between the neighboring nanotubes. Note the non-monotone dependence of composite conductivity on the CNTs content.

It is of interest to compare the current-voltage characteristics obtained within the frame of the above-described tunneling model with experimental data. However, the direct comparison is rather difficult, because the magnitudes of necessary parameters are not given in the above-cited publications. Besides that, the CNTs content in the experiments considerably exceeds the percolation threshold, so that the charge transport in these conditions proceeds not through a 1D channel but rather through a branched percolation cluster. Since the length and diameter of CNTs comprising such a cluster are not known with reasonable accuracy, it is hardly possible to make the right conclusion on the number of contacts between CNTs, and therefore to calculate the electric conductivity of such a cluster even within the frame of the simple model. Nevertheless, Figure 8 demonstrates a qualitative agreement between calculated dependences of the contact circuit resistance on the applied voltage (5) and measured data [14,15]. These dependences are expressed in relative units, so that *R*_m_ (*U*_m_) is the maximum value of resistance. A good agreement between the experimental dependences [15] and calculation results (5) occurs at *μ* = 1 nm and *σ* = 0.1 nm. Experimental dependences [14] match the calculation results (5) at *μ* = 0.6 nm and *σ* = 0.06 nm. A good agreement between the measured and calculated dependences demonstrates the applicability of the model used.

### 2.3. Dependence of the Contact Resistance on the Intersection Angle

It has been shown above that the main reason of non-Ohmic behavior of polymer nanocomposites doped with CNTs is the non-perfect contacts, the resistance of which depends drastically of both inter-tube distance and applied voltage. Besides that, the contact resistance also depends on the intersection angle between nanotubes. This issue has been studied in detail theoretically by the authors [26]. Two intersecting CNTs with the chirality indices (5, 5), containing 280 atoms were considered. The angle between the nanotubes was varied between *φ* = 0 (parallel configuration) and *φ* = π/2 (perpendicular configuration). The distance between the nanotubes was set to 3.4 Ǻ and 6.0 Ǻ. In the case of parallel configuration, the nanotubes overlap by a half of their length. The electron structure was calculated with the use of code [27] based on ab initio approach developed in [28,29]. The dependence of the probability of under barrier electron propagation on the electrical field was calculated on the basis of the above-mentioned electronic structure with the use of some modification of the Green function method for various angles of mutual orientation of nanotubes and different inter-tube distances. These dependences have been utilized for calculation of the contact CVCs. The results of the calculations are presented on Figure 9. As is seen, the dependence of the contact resistance on the mutual orientation angle determines a non-linear character of the CVCs. One can assume that in a real situation, a superposition of the two above-described effects occurs. However, the former effect seems to be more essential, regarding a qualitative agreement with experiments (see Figure 8).

### 2.4. Polymer-Based Composites with Graphene Filling

Currently, along with CNTs, one- or few-layer graphene fragments are also used as a filler of polymer-based composites. This changes the properties of such composites and notably expands the field of their application. Thus, the authors of [30] have fabricated and studied epoxy-based composites doped with graphene flakes and copper nanoparticles. Both few-layer graphene flakes and copper nanoparticles were of several μm in size. An abrupt increase of the thermal conductivity of the composite has been observed, which reached the value of 13.5 ± 1.6 Wm^−1^K^−1^ at loading 40 wt% and 35 wt% copper nanoparticles, while the thermal conductivity of the initial epoxy matrix is as low as 0.2 Wm^−1^ K^−1^. Besides that, the authors report on the synergy effect, so that the increase of thermal conductivity due to inserting two fillers notably exceeds that calculated as a sum of two increments. The synergy effect was also observed by the authors [31] who fabricated and studied epoxy composites with graphene and boron nitride fillers. Filling epoxy matrix with graphene and nitride boron in different content allows one to produce composite with desired thermal and electric conductivity. The graphene flakes and BN particles to be filled were of similar size: The thickness ranged from 0.35 to 12 nm in and the lateral size ranged from 2 to 8 μm. The measurements performed indicate a monotone increase of the thermal conductivity of a composite with enhancement of filler loading. At fixed filler loading, the thermal conductivity increases linearly with the enhancement of the graphene fraction in the filler. The maximum value of the thermal conductivity (about 8 Wm^−1^ K^−1^) is reached at a total filler loading of 43.6 vol.% and the content of graphene of 100%.

Inserting graphene flakes into a polymer matrix allows the composite to absorb the electromagnetic radiation. This effect has been demonstrated by the authors of recent publications [32,33,34,35]. Epoxy-based composites with only 1 wt% single- or few-layer graphene fillers demonstrate very low electromagnetic transmission T < 1% for samples of about 1 mm in thickness in the extremely high-frequency electromagnetic field (220–325 GHz) [35]. The composite samples of 1 mm in thickness and a graphene loading of 8 wt% provide an electromagnetic shielding of 70 dB in the sub-terahertz frequency band with negligible energy reflection to the environment. Composites of the basis of poly(lactic) acid doped with carbon-based nanostructures—multiwalled carbon nanotubes (MWCNTs) and graphene nanoplatelets (GNPs)—with the content up to 6 wt% were produced and studied in terms of the electromagnetic shielding properties by the authors [33]. It was found that the electromagnetic shielding efficiency (EMI) of nanocomposites strongly depended on the aspect ratio of the nanofillers, whereas the type of processing technique did not have a significant effect. The authors [34] indicated a notable influence of annealing epoxy resin composites filled with GNP on their electromagnetic properties. The annealing temperature reached 500 K. As a result of annealing, the dc conductivity of epoxy matrix filled with 4 wt% GNP was enhanced by 68 times. Besides that, the annealing substantially lowers the percolation threshold, from 2.3 wt% for as-produced samples to 1.4 wt%. For a given GNP concentration, the tunnel barrier decreases after annealing. One should note that graphene-doped composites can absorb the electromagnetic radiation even below the electrical percolation threshold, so that the absorbing composite remains electrically insulating [32]. The graphene-doped composites can be used as electromagnetic absorbers in the high-frequency microwave radio relays, microwave remote sensors, millimeter wave scanners, and wireless local area networks.

Measurements [36] indicate the aging of polymer-based composites with nanocarbon filling. The authors studied polylactic acid filled with 12 wt% of graphene nanoplatelets (GNP), multiwall carbon nanotubes (MWCNT), and their hybrid mixture (1:1). It has been found that solid annealing at 80 °C for 4 h and pre-melt annealing at 120 °C for 3 h enhance the crystallinity, thermal stability, electrical conductivity, and tensile Young’s modulus, along with reduced tensile strength, elongation, and toughness, compared to the neat polymer.

## 3. Percolation Behavior of Reduce Graphene Oxide

### 3.1. Chemical Reduction of Graphene Oxide

One more example of carbon-based nanocomposite is presented by reduced graphene oxide (rGO) (see [5,6,19,37,38,39,40,41,42,43] and works cited there). This material consisted of flakes of partially reduced graphene oxide. Electrical propertied of such a material are determined by the reduction degree [6].

The most widespread approach to the reduction of GO is based on the use of chemical reagents [37,38,39,40,41,42,43]. An example of the procedure of chemical reduction of GO has been presented by the authors of [43], who used as an initial material a water suspension GO (concentration 2 g/L) produced by the standard Hammers method [44]. In accordance with this method, 2 g graphite powder was added to 50 mL sulfuric acid (98%). After stirring for several minutes, 6 g KMnO_4_ was added to the suspension. Then, the mixture was heated up to 35 °C and diluted with 100 mL water. The suspension obtained was kept for 30 min at a temperature of 70 °C; thereafter, 100 mg more water was added. After that, 10–15 mL 3% water solution hydrogen peroxide was inserted into the suspension, which make the mixture colorless. This mixture passed through a filter paper, and the material that remained on the filter was flooded with 600 mL distilled water and stirred for 10 min by means of a magnet mixer. GO flaxes were separated from other reaction products by centrifugation. Then, 100 mL of suspension produced was mixed with 50 mL hydrazine hydrate, ultrasonicated, and deposited onto a glass or silicon substrate, which was dried at a room temperature. Another part of the suspension (200 mL) was dried in a Petri cup for 24 h at a temperature of 100 °C, which resulted in the formation of a black paper-like material of 20–30 μm thick.

The CVC of a reduced GO sample of 3 × 1.5 cm^2^ in size was measured by means of the four-contact method. The results of the measurement are presented on Figure 10. As is seen, the material demonstrates a non-Ohmic behavior inherent to percolation systems.

### 3.2. Thermal Reduction of GO

The procedure of chemical reduction of GO has a drawback related to the usage toxic reagents such as hydrazine and hydrazine hydrate. This drawback hinders a wide spread of the chemical reduction GO method and stimulates developing alternative approaches. The most natural approach to the reduction of GO avoiding the above-mentioned drawback is the thermal processing of a material resulting in splitting out radicals such as oxygen, СО, ОН etc., from GO fragments [5,6,7,19,45,46,47,48,49,50]. The binding energy of such radicals with graphene structure is several times lower than that required for destroying of such a structure therefore it is possible to find the conditions at which splitting out radicals is not accompanied by decomposition of graphene lattice. Results of the thermal reduction of GO are very sensitive to parameters of the process (evolution of the temperature, duration of the process, composition, pressure and flowing velocity of buffer gases, and so on), therefore efforts of many groups are focused towards the optimization of this procedure. Thus, the dynamics of thermal reduction of GO was elaborated by means of the thermogravimetry analysis (TGA) permitting determination of the rate of the weight loss of a sample as a function of the annealing temperature [46]. Measurements indicate that at heating a sample with a rate of 10 °C/min a notable loss in the weight of a sample occurs at a temperature of 180 °C. The combination of thermogravimetry with mass-spectrometry measurements has permitted one to conclude that the main gases evolved at heating GO are CO, H_2_O, and CO_2_. It has been found that the evolution of H_2_O occurs at all the temperatures while CO_2_ evolves beginning from 130 °C. The measurements performed by the differential scanning calorimetry permitted the determination of the activation energy of the reduction process which is equal to 1.73 eV. The content of radicals –ОН and –C was It is OK determined basing on processing X-ray photoelectron spectra (XPES) of the samples. Figure 11 presents the results of such processing. As is seen, splitting out OH radical already occurs at a temperature of 90 °C.

The authors [46] studied, in detail, the evolution of electron characteristics of GO in process of thermal reduction. The water suspension of GO synthesized by the standard Hummers method was dried. The powder obtained was subjected for an hour to thermal processing in a cylindrical chamber of a furnace at N_2_ atmosphere at temperatures of 250, 300, 350, 400, 500, 700, 800, or 1000 °C. The furnace was heated with a rate of 5 °C/min, and nitrogen was pumped through with a rate of 50 mL/min. Vacuum filtration of the suspension of partially reduced GO in N,N-dimethylformamide (DMF) resulted in the deposition of reduced GO (rGO) films on the filters. The films were annealed for an hour under N_2_ flow at a temperature of 150 °C with the aim of removal adsorbed water and oxygen molecules. The concentration of charge carriers and their mobility were determined by the standard Hall measurements in dependence on the annealing temperature. The sign of the current carriers was established basing on the measurements of the Seebeck coefficient. Results of the treatment of these measurements are given in Table 1. One should note the change of the current carriers’ sign as the annealing temperature changes. The authors [46] assign this effect to the influence of functional groups on the character of the charge transport. One should note that the above-mentioned result has not be confirmed in subsequent publications. Nevertheless, the results of [46] indicate that the set of samples GO annealed at different temperatures can be considered as a new class of semiconducting materials with a wide range of electronic characteristics.

### 3.3. Electrical Characteristics of rGO

The procedure of thermal reduction of GO usually consists of heating a sample up to temperature of several hundred °C for several minutes or tens of minutes. However, the reduction procedure can be performed at essentially lower temperatures, which requires much more extended time. Thus, authors [50] performed the thermal treatment of a water suspension GO at temperatures 50 and 80 °C with the duration of processing from one to nine days. Drying the suspension resulted in a set of samples with different duration of processing. Four-contact measurements indicate that the enhancement of the processing duration is accompanied by a considerable lowering in the resistance of the material. Figure 12 presents the results of these measurements. As is seen, the resistance of the GO sample annealed for six days at a temperature of 80 °C decreases by more than four orders of magnitude.

The electric conductivity is a very convenient indicator of the degree of GO reduction. This parameter was used in [6,19] as a major number determining the dynamics of transformation of GO to a state close to graphene. Graphene oxide produced by the Hummers method [44], with utilization of potassium nitrate, potassium permanganate, and concentrated sulfur acid, was used as an initial material. Paper-like sheets of GO produced by such a method were 40–60 μm in thick and 1.2 g/cm^3^ in density. These sheets were cut into rectangular stripes of 10–15 mm in width and 15–25 mm in length and experienced a thermal treatment in the high-temperature furnace planarGROW-2S produced by the PlanarTech (The Woodlands, TX, USA) company for CVD synthesis. The regime of the treatment is pregiven by a computer that controls all the parameters of the procedure. The thermal treatment is performed in a cylindrical tube of 40 cm in diameter. The samples are placed into a quartz boat of 20 cm in length, 3 cm in width, and 2.5 cm in depth, which is inserted into the tube. Heating the furnace and thermal processing of the samples were performed at a slow pumping Ar with the rate of 50 cm^3^/min (reduced to normal conditions) at Ar pressure of 10 Torr. Experiments performed show that the reliable, well-reproducible results can be obtained only at a rather low rate of heating the samples up to desired temperature. Thus, heating samples with the rate higher than 1 °C/s results in non-controllable explosion-like destroy of the material. For this reason, the rate of heating the furnace from a room temperature up to 200 °C was 2 °C/min, while the rate of the subsequent heating up to the annealing temperature was ~1 °C/s. The duration of the thermal treatment was 10 min at all the temperatures. After termination of the annealing, the furnace is switched out, and its cooling down to room temperature in a natural manner took about one hour.

The CVCs of the samples annealed at different temperatures were measured by the usage of a standard electrical apparatus. A sample was clamped between the contacts by means of cupper foil braces, which provided a homogeneous passage of the current through all the film. The dynamics of changing the density of samples in the process of their annealing was measured with the usage of the electronic balance Sartorius QUINTIX124. The size of the samples was measured by a standard micrometer. Measurements indicate that the non-homogeneity of the samples by the depth accounted for about 20%. Since the depth of samples is used at determination of the conductivity of the material, this non-homogeneity is the main source of the measurement error. The results of the measurements are given in Figure 13.

Figure 14 presents results of measuring the conductivity of samples as a function of the temperature. As is seen, a sharp increase in the conductivity of samples (about by five orders of magnitude, from 10^−3^ up to 100 S/m) occurs within rather narrow range of the annealing temperature (several degrees). The position of this range depends on the annealing duration and shifts toward lower temperature as the duration increases. One should note that the jump in the conductivity at changing the annealing temperature in extremely sharp so that the authors [6,19] have not managed to obtain the samples with the conductivity within the above-mentioned range. This indicates a percolation mechanism of the conduction, in accordance with which a material transforms from the isolating to the conducting state as a result of the formation of one or a few conducting paths formed by a chain of contacting fragments of reduced GO.

Further annealing the material results in a smoother enhancement of the conductivity. The maximum reached value of the conductivity of the reduced GO (~3500 S/m) is about an order of magnitude lower than the reference value for graphite. However, taking into account that the density of the material annealed at a temperature of 800 °C accounts for about 0.5 g/cm^3^ (see Figure 13), which is about 4.5 times lower than that of graphite, one can conclude that the conductivity of the material accounting for one graphene layer is only twice lower than that for graphite. Since the density of the sample annealed at a temperature of 800 °C is 4.5 times as low as that for graphite, one can believe that the average distance between layers in such a material exceeds the corresponding value for graphite (0.34 nm) by the same factor. Therefore, this material presents a layer structure with an average inter-layer distance of about 1.5 nm and the conductivity (accounted for one layer) close to that for graphite. At such a distance, the interaction between the neighboring layers is negligible so that it is natural to conclude that the layers involved in such a structure are close to graphene in their characteristics. Annealing results in the thermal reduction of GO fragments, which lose added oxygen and transform to a conducting state. The conductivity of such a material is determined by the resistance of contacts between those fragments, which decreases as the applied voltage enhances.

### 3.4. XPS Spectra of Reduced GO and Plasmons

The analysis of XPS spectra of partially reduced GO permits one to determine the change of electronic characteristics of the material in the process of its annealing [51,52]. The spectra of GO samples annealed at various temperatures were measured using the apparatus Kratos Axis Ultra DLD. The survey spectra were measured in the transmission regime at *E*_pass_ = 160 eV, while the electron energy loss spectra in the energy region adjacent to C 1s peak—at *E*_pass_ = 40 eV. Four samples were loaded to the holder simultaneously. The spectra were produced by the use of monochromatic line Al Kα. The spectra of samples annealed at a temperature of 200 °C and lower were measured using a neutralizer because of their charging. The samples annealed at higher temperatures had sufficient conductivity, so that the neutralizer was not used at measuring XPS spectra. The results of measurements are given on Figure 15.

The measured spectra were exposed to the mathematical treatment taking into account multiple elastic and inelastic processes of electron scattering on atoms of the target, instrumental function of the energy analyzer, and Doppler line broadening with the use of the methods of solution of similar problems developed before [53,54,55]. Table 2 presents the data on the chemical composition of reduced GO depending on the annealing temperature [19]. These data have been obtained in result of processing XPS spectra. As is seen, an increase in the annealing temperature results in a decrease in the oxygen content. Oxygen was not found in samples annealed at 1000 °C.

A distinctive peculiarity of the XPS spectra of reduced GO is the appearance of the peak caused by the interaction of electrons with plasmon oscillations (plasmon peak). This peak occurs at annealing temperatures exceeding 200 °C, and its intensity increases as the annealing temperature enhances. The appearance and the evolution of the plasmon peak relate to the dependence of the conductivity of reduced GO on the annealing temperature (Figure 14). As a result of the percolation transition, the conductivity of a sample increases by several orders of magnitude, which is accompanied by the appearance of plasmon oscillations. Figure 16 presents the comparison of dependences of the plasmon peak intensity and the conductivity of reduced GO on the annealing temperature [51]. The similarity of these dependences indicates a physical correspondence of the reduced GO conductivity and the plasmon peak intensity. One should note that the existence of the plasmon oscillations in partially reduced GO offers a possibility for the usage of this material for amplification of Raman signal. Such a possibility, demonstrated recently for carbon nanotubes [56], seems to be very attractive because of the miniature size and relatively high affordability of GO samples.

### 3.5. Non-Ohmic Conduction of Reduced GO

Partially reduced GO is consisted of single-layer fragments (flakes) of several μm in size. At a low reduction degree corresponding to relatively low values of the annealing temperature such fragments do not conduct electricity, so that GO samples demonstrate the properties of an insulator. As the temperature and, correspondingly the reduction degree increase, the conductivity of such fragments enhances. Conducting GO flakes come into contact with each other and form percolation conducting channels at exceeding some critical value of the reduction degree. The resistivity of the material decreases by several orders of magnitude. At a low reduction degree, the resistance of fragments exceeds that of contacts, so that the resistivity of the composite is determined by the resistance of fragments and does not practically depend on the applied voltage. A rise in the reduction degree of GO fragments is accompanied by a decrease of their resistance, and at quite a high reduction degree it becomes (on average) lower than the characteristic contact resistance. In such conditions, the material can be considered as a chain of contacts distinguishing in the distance between fragments. The resistance of such a chain depends notably of the applied voltage, like it occurs in the above-considered case of polymer nanocomposites doped with CNTs (see Section 2.1). Therewith a GO sample annealed at quite high temperatures demonstrates a non-Ohmic behavior, in accordance with which the conductivity of the material increases with the rise of the applied voltage.

Figure 17 presents the current-voltage characteristics of GO annealed at various temperatures [19]. Treatment of these data results in a dependence of the resistivity of GO samples processed at different temperatures on the applied voltage. These dependences are shown on Figure 18 [19]. As is seen, a practically linear rise of the conductivity with an increase of the applied voltage is observed for all the samples, independently on the annealing temperature. Such a behavior indicates a percolation character of the conduction of the material, in accordance with which the charge transport is carried out through a limited number of paths formed by contacting GO fragments. In such conditions the conductivity of the material is determined mainly by the contact resistance, which, in turn, is caused by the quantum tunneling of electrons through the barrier between the neighboring conducting GO fragments. An increasing dependence of the tunneling probability on the applied electrical field results in decreasing the resistance of the sample as the applied voltage enhances. Therefore, GO samples demonstrate non-Ohmic conduction that has had been noted recently for polymer composites doped with CNTs [5,8,9,10,11,12,13,14,15,16,17,18,19] (see Section 2 of this article). The degree of dependence of the conductivity on the applied voltage is determined by the annealing temperature.

The dependence of the conductivity of a sample *σ* on the applied voltage *U* is close to a linear one, so that it is convenient to express it by the following relation
(10)$$\sigma =\sigma_{o}\left[1+k(U-U_{o}\right]$$
where *σ*_o_ is the conductivity of the sample at some voltage *U*_o_, *k* is the empirical parameter presenting the derivation of the conductivity over the applied voltage *k* = d*σ*/d*U.*
Figure 19 shows the dependence of this parameter on the annealing temperature. In accordance with the above-given analysis, the degree of dependence of the conductivity of reduced GO on the applied voltage increases as the annealing temperature and, therefore, the GO reduction degree enhances.

### 3.6. Model Description on the Non-Ohmic Conduction of GO Samples

The theoretical determination of the conductivity of a material consisted of a set of reduced GO fragments being in non-ideal contact with each other requires for evaluation of the conductivity of a percolation chain with taking into account electron tunneling through contacts having different characteristics [1,19]. This problem seems to be extremely complicated in the technical aspect due to uncertainty of the structure of reduced GO fragments and a spread of remove the reduction degree and, correspondingly, conductivity. However, some qualitative conclusions on the dependence of the conductivity of such a material on the applied voltage can be made within the frame of a relatively simple model approach, in accordance with which a sample is considered as a system consisting of two fragments of reduced GO contacting with each other in non-ideal manner. The character of charge transport through such a contact is determined by electron tunneling through the potential barrier separating the neighboring reduced GO fragments. The resistance of such a model object is expressed by the following relation
(11)$$R=2R_{f}+R_{c}$$
where *R*_f_ and *R*_c_ is the resistance of the object and contact, correspondingly. The conductance of the object under consideration *σ* is inversely proportional to its resistance:(12)$$\sigma ∼\frac{1}{2R_{f}+R_{c}}$$

The resistance of the reduced GO fragment is characterized by a decreasing dependence on the annealing temperature. The character of this dependence can be seen from Figure 14 presenting the dependence of the conductivity of the material on the treatment temperature. As is seen, the resistivity of GO changes by more than six orders of magnitude within the temperature range where the measurements were done. One can suppose that at relatively low annealing temperatures (lower or of the order of 200 °C) the reduction degree is rather low, so that *R*_f_ >> *R*_c_, while at temperatures exceeding 500 °C the inversed relation takes place. Therefore, in accordance with (12), in a low-temperature region, the conductance of the model object under consideration is determined by the resistance of reduced GO fragments that does not depend on the applied voltage. In this range, the value of the parameter *k* involved into (10) is close to zero. At temperatures exceeding 500 °C the conductance of GO fragments is quite high and exceeds the characteristic value of the contact conductance that depends drastically on the applied voltage and does not depend on the annealing temperature. In this temperature range, the parameter *k* takes the maximum value that does not practically depend of the annealing temperature. In an intermediate temperature range of 200 ≤ *T* ≤ 500 °C, the resistance of the GO fragment is of the same order as contact resistance *R*_f_~*R*_c_. In this range, the conductance of the model object depends notably on the applied voltage, and the degree of such a dependence (parameter *k*) increases with the enhancement of the annealing temperature.

Let us estimate the dependence of the contact resistance *R*_c_ on the applied voltage. One should take into account that the probability of electron tunneling through the barrier formed in a vicinity of non-ideal contact between the neighboring conducting GO fragments depends, in an exponential manner, on the distance between these fragments. Applying an external electrical field results in a change of the barrier shape so that the probability of passing the barrier increases as the electric field enhances. This causes the dependence of the conductance of the model object under consideration on the electrical field strength. Analyzing the dependence of the contact resistance between two neighboring GO fragments on the distance and the potential difference between those, we will suppose that the conducting electrons are found in 1D rectangular potential well the depth of which is determined by the electron work function *φ*_e_. The value of this parameter is about 5 eV, which exceeds the characteristic value of the kinetic energy of conducting electrons (*E*_e_~300 K) by more than two orders of magnitude. Therewith the electron transport from one GO fragment to another one proceeds as a result of tunneling through a rectangular barrier of *φ*_e_—*E*_e_ in height. Applying an electrical field *F* changes the shape of the barrier, so that the probability of electron tunneling is expressed through barrier parameters and the electrical field strength by the relation (2), in accordance with the known result of the quasi-classical approximation [22]. This results in the relation (3) for the dependence of the contact resistance on parameters *d* and *F*. When the percolation threshold is only slightly exceeded, the charge transport proceeds by a single or few percolation paths. In this case, the resistance of one channel is expressed through the sum of resistances of fragments and contacts between those (4). However, in distinction on the above-considered polymer nanocomposites doped with CNTs, in the case of partially reduced GO with a low reduction degree, one cannot believe that the resistance of GO fragments exceeds a typical contact resistance. As the reduction degree enhances, the fragment resistance decreases, so that the inter-relation between two these values of the resistance changes.

The dependence of the parameter *k* defined by the relation (10) on the annealing temperature can be estimated within the frame of the above-described model, in accordance with which a sample consists of two reduced GO fragments separated by a non-perfect contact. This model takes into account that at low annealing temperatures, when *R*_f_ > *R*_c_, the conductance of the sample is determined by that of GO fragments, while at higher temperatures, when the sample conductance is the inversed inequality occurs, the sample’s conductance is determined by the contact resistance, which is very sensitive to the magnitude of the applied voltage. The dependence of the parameter *k* on the annealing temperature was calculated on the basis of relation (12), where the experimental dependence of the GO fragment resistance *R*_f_ on the treatment temperature shown on Figure 14 and the dependence of the contact resistance *R*_c_ on the applied voltage (3) were used. The calculation results are depicted on Figure 19 by solid line. The calculated and experimental values *k* coincide closely in the regions of low and high annealing temperature. Some discrepancy between these values in an intermediate temperature range can be explained by a simplicity of the model used, which considers a sample consisting of two GO fragments and does not take into account a spread in the distance between neighboring GO flakes. Such a spread results in lowering the parameter *k* because the current flows mainly through contacts with minimum inter-flake distance. A qualitative agreement between the experimental and calculated dependences confirms an applicability of the conduction model used.

The model developed earlier for description of the percolation conduction in polymer nanocomposites doped with CNTs (see Section 2) was used for calculation of electric characteristics of GO samples. According to this model, the charge transport is performed by a percolation path formed by reduced GO fragments contacting with each other in a non-perfect manner. It is assumed that the conductance of GO fragments exceeds that of contacts so that conductance of the percolation channel is determined by the contact resistance which in its turn depends on the applied voltage. The resistance of the percolation channel is expressed through the sum of contact resistances by the relation (5). Therewith the normal distribution (8) of contacts by inter-fragment distances is supposed. The CVCs of percolation channels of various length *n* for different values of parameters *μ* and *σ,* determining the mean inter-fragment distance and its standard deviation, correspondingly, were calculated following the approach described in Section 2 of the present article. The set of distances *d*_1_, *d*_2_, …*d*_n_ was created by means of the random number generator. The resistance was calculated repeatedly using different sets of values *d*_1_, *d*_2_, …*d*_n_ for each chosen pair of parameters *μ* and *σ*. The calculation results are presented on Figure 20 [21].

A direct comparison of dependences given on Figure 20 with those obtained experimentally is rather difficult due the lack of the necessary information on the geometry of fragments comprising the percolation chain. However, one can find the conditions under which the measured CVC of GO samples correspond to the results of the calculations performed. Such a comparison has been presented on Figure 21. As is seen, the best agreement between the calculated and measured dependences is reached at different values of the mean inter-fragment distance, depending on the annealing temperature. As the annealing temperature increases, the mean inter-fragment distance *µ* providing a good agreement with the experiment decreases. This can be considered as an indication of changing the structure of the percolation channel: The content of conducting fragments enhances, which results in an increase of the contribution of contacts with lower inter-fragment distances.

## 4. Conclusions

Carbon nanocomposites can be treated as a non-conducting matrix doped with conducting nanoparticles. Two types of such composites are analyzed in the present article: polymer composites doped with CNTs and partially reduced graphene oxide. In the composites belonging to the first type, the role of conducting particles is played by carbon nanotubes, while for partially reduced GO this role is played by fragments of reduced GO. At low content of conducting carbon nanoparticles the charge transport in such composites is performed through one or a few percolation paths formed by nanoparticles contacting with each other. Most of these contacts are not perfect in typical conditions. It is caused by a sharp dependence of the probability of electron tunneling through the barrier formed by these contacts in the presence of the electric field, on the barrier’s width. Such a peculiarity of the materials results in non-Ohmic behavior of nanocomposites, which manifests itself in an increasing dependence of the conductivity on the applied voltage. Model calculations performed for both types of nanocomposites result in non-linear current-voltage characteristic of the material, which agree qualitatively with experimental data. In the case of thermally reduced GO, the degree of non-linearity is determined by the reduction degree, which in turn relates to the annealing temperature. Therewith in the process of the thermal reduction of GO, a new type of the percolation transition is observed. According to this, percolation paths are formed not as a result of an enhancement of the quantity of conducting particles but as a result of the increase in the conductivity of GO particles due to the thermal treatment. The percolation transition is accompanied by a jump in the conductivity by 5 orders of magnitude within the temperature range between 150 and 180 °C. The transition proceeds very sharply, so that the accurate value of the transition temperature could not be determined.

## Figures and Tables

**Figure 1 ijms-21-07634-f001:**
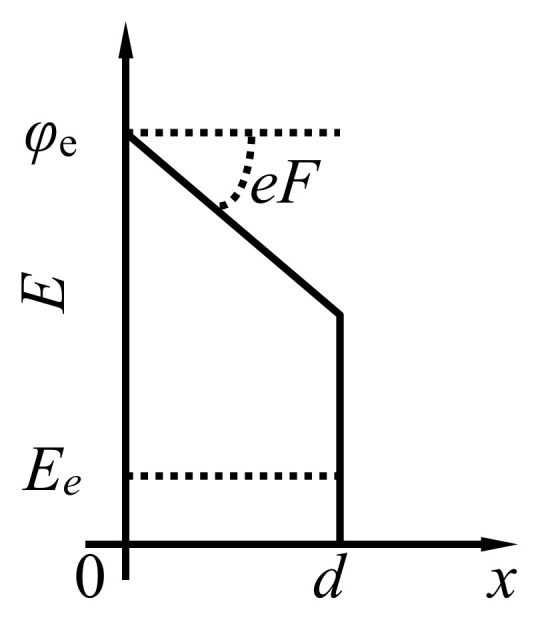
The shape of the potential barrier separating the electrons belonging to the contacting nanotubes.

**Figure 2 ijms-21-07634-f002:**
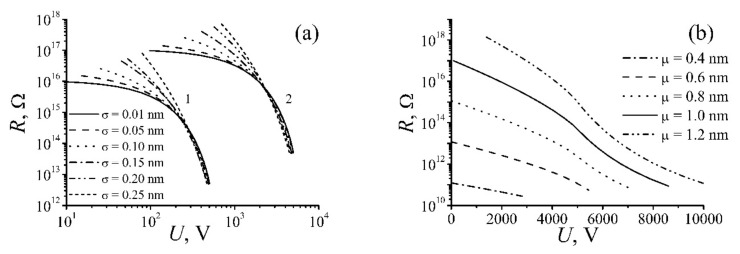
Dependences of the resistance of a percolation channel calculated for different parameters of the chain: (**a**) *μ* = 1 nm; the number of contacts *n* = 100 (**1**) and 1000 (**2**); (**b**) *σ* = 0.01 nm, the number of contacts *n* = 1000 [21].

**Figure 3 ijms-21-07634-f003:**
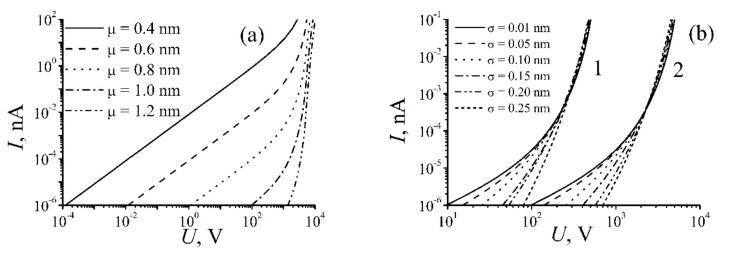
CVC of a percolation chain calculated for various values of parameters *σ*, *μ*, and *n*: (**a**) *σ* = 0.01 nm, the number of contacts *n* = 1000; (**b**) *μ* = 1 nm, *n* = 100 (**1**) and 1000 (**2**) [21].

**Figure 4 ijms-21-07634-f004:**
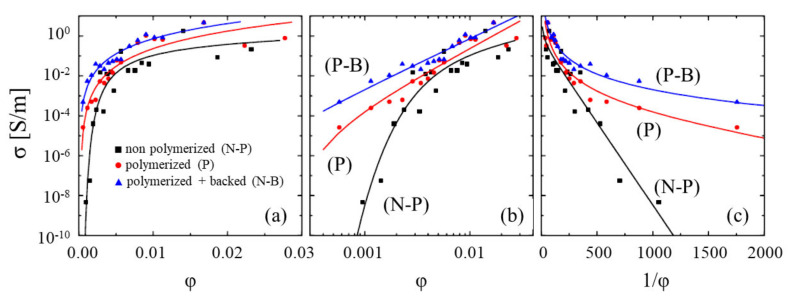
(**a**) Results of measurements of the dependence of the conductivity of various types CNT-SU8 nanocomposite on CNT weight concentration; (**b**) the same dependence represented in a double logarithm scale; (**c**) dependences of logσ on 1/φ. The linear character of the dependence indicates the role of tunneling in the electron transport process [15].

**Figure 5 ijms-21-07634-f005:**
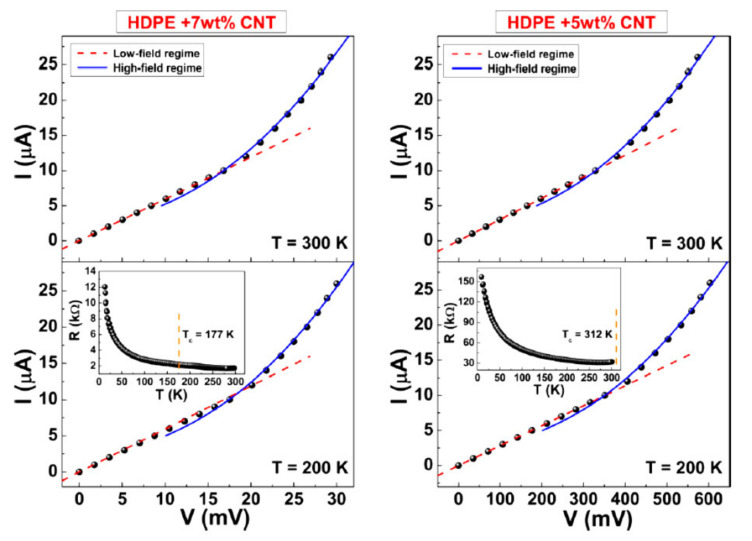
CVCs of samples of composite HDPE+7% carbon nanotubes (CNTs) (**left**) and HDPE+5% CNTs (**right**) measured at temperatures 300 and 200 K. Dotted and solid lines reflect approximation dependences at low and voltages. Temperature dependences of the resistance are shown on insets [14].

**Figure 6 ijms-21-07634-f006:**
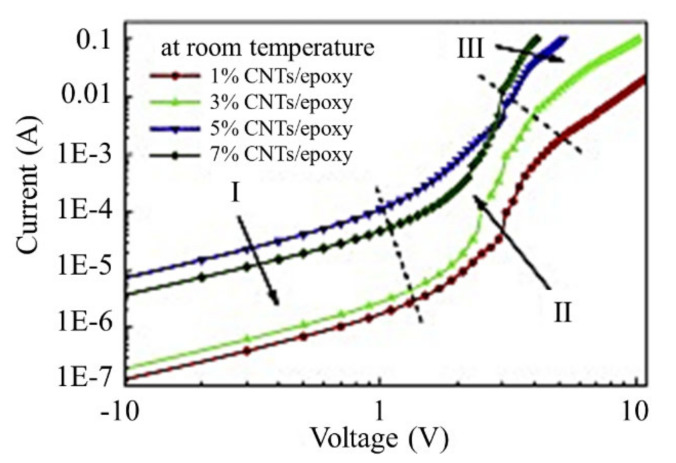
CVCs of nanocomposite on the basis of epoxy resin doped with CNTs measured for samples with various content of additive [9].

**Figure 7 ijms-21-07634-f007:**
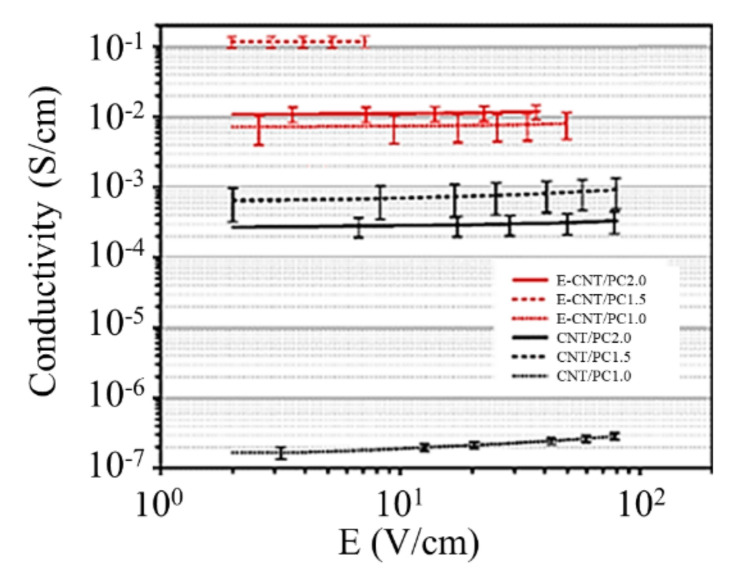
Dependences of the conductivity on the electric field measured for polycarbonate samples doped with CNTs with (E-CNT/PC) and without (CNT/PC) conductive polymer coverage [17]. The mass content of the dopant (%) is shown on the picture.

**Figure 8 ijms-21-07634-f008:**
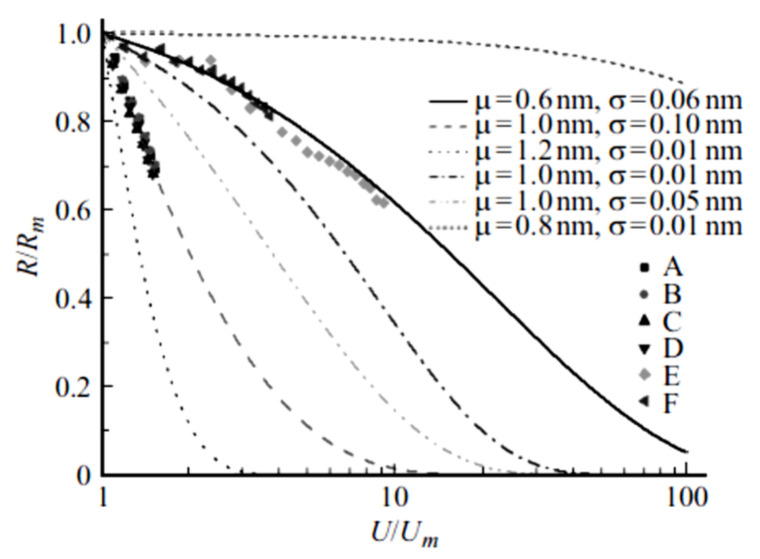
Comparison of calculated and measured [14,15] dependences of resistivity of nanocomposites doped with CNTs on the applied voltage (relative units). Calculations: The number of contacts *n* = 1000, the values *μ* and *σ* are shown on the picture. Experimental points: (**A**)—high-pressure polyethylene (HPPE) + 7% (weight) multi-walled CNTs, T = 300 K; (**B**)—HPPE + 5% CNTs, T = 300 K; (**C**)—HPPE + 7% CNTs, T = 200 K; (**D**)—HPPE + 5% CNT, T = 200 K [15]; (**E**)—polyvinilbutiral (PVB) + 3% CNTs; (**F**)—polydimethylsiloxane (PDMS) + 1% CNTs [14].

**Figure 9 ijms-21-07634-f009:**
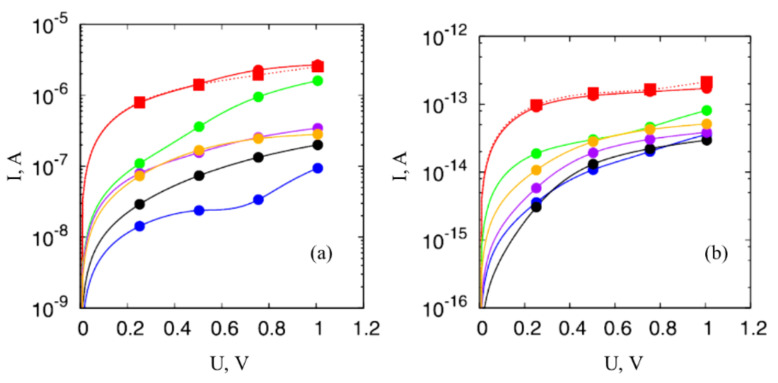
CVCs of contact between two nanotubes with chirality indices (5,5), calculated for different angles between CNT axes; (**a**): Inter-tube distance is 3.4 Ǻ, (**b**): 6.0 Ǻ. Red lines correspond to *φ* = 0, green: *φ* = 0.1π, blue: *φ* = 0.3π, black: *φ* = 0.3π, violet: *φ* = 0.4 π, orange: φ = 0.5π. Solid and dotted lines correspond to calculations for different inter-atom interaction potential [26].

**Figure 10 ijms-21-07634-f010:**
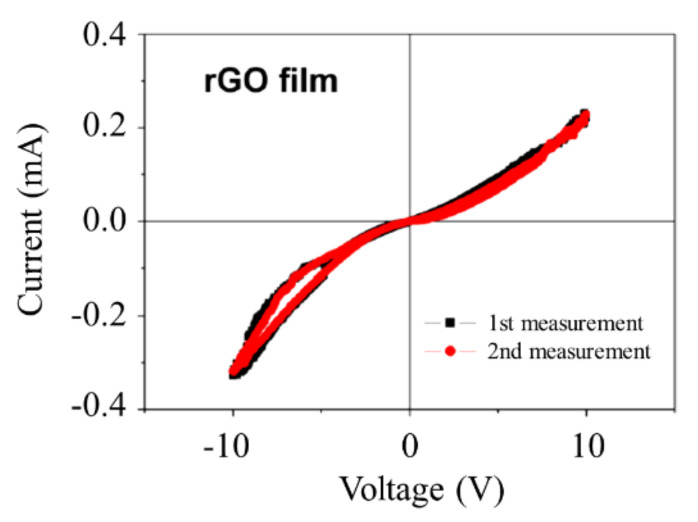
CVC of a reduced graphene oxide (GO) film [35].

**Figure 11 ijms-21-07634-f011:**
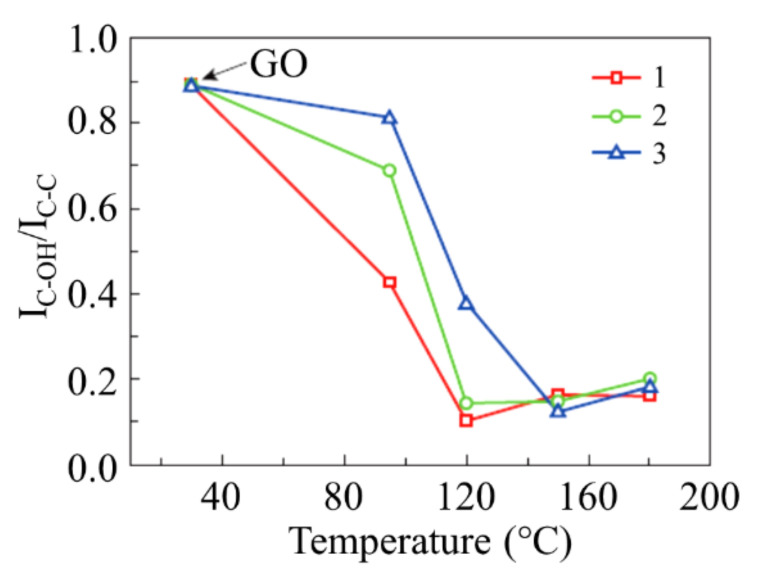
The ratio of intensities of peaks C-OH and C-C as a function of the temperature determined basing on processing XPES spectra of samples: (**1**) The sample is immersed into deionized water; (**2**) the sample is situated in a sealed off quartz tube at a pressure of few Pa; (**3**) the sample is found on an open air [46].

**Figure 12 ijms-21-07634-f012:**
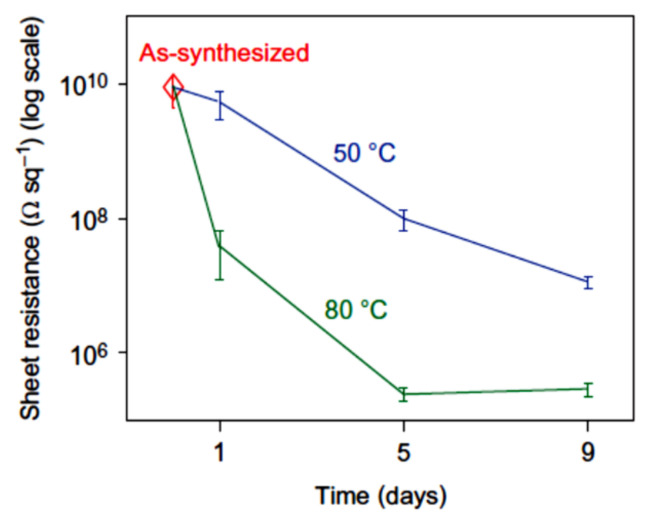
Dependences of the resistance of thin films GO on the duration of annealing measured at various temperatures [50].

**Figure 13 ijms-21-07634-f013:**
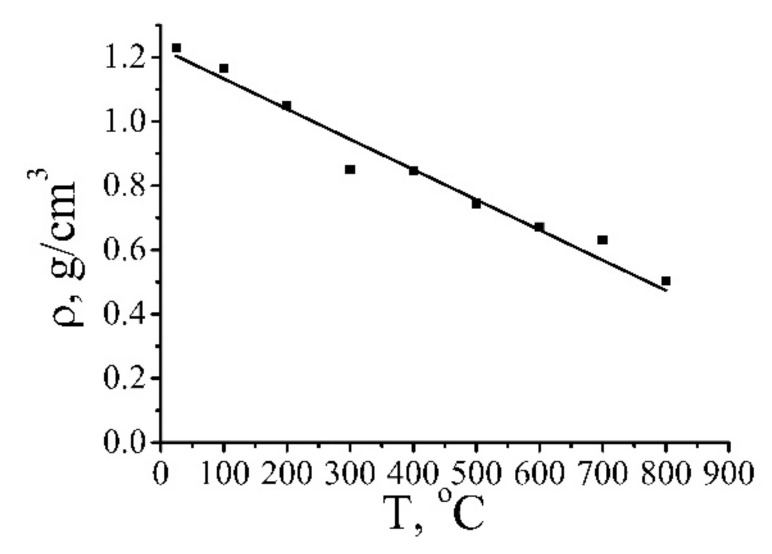
Dependence of the density of partially reduced GO on the annealing temperature [6,19].

**Figure 14 ijms-21-07634-f014:**
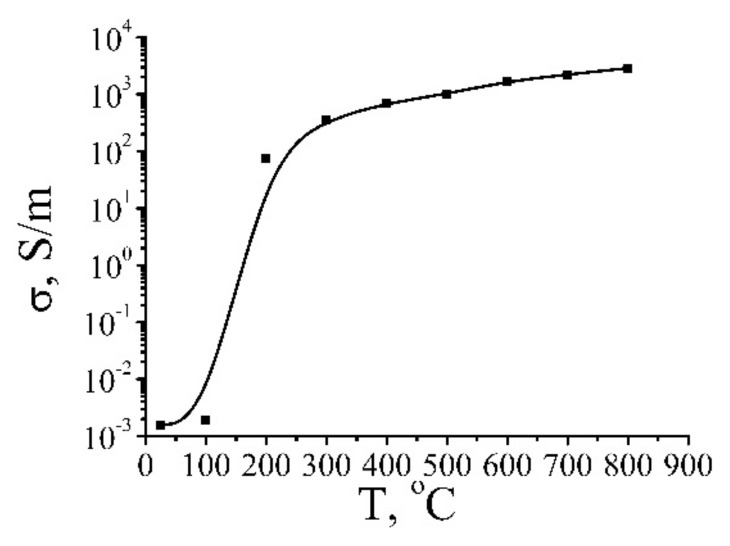
Dependence of the conductivity of reduced GO samples on the annealing temperature. The points have been obtained in result of averaging over many samples and several magnitudes of the applied voltage [6,19].

**Figure 15 ijms-21-07634-f015:**
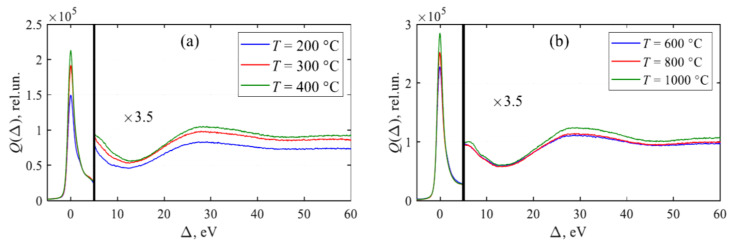
XPS spectra of GO samples annealed at various temperatures. Δ = BEC_1s_-E [51].

**Figure 16 ijms-21-07634-f016:**
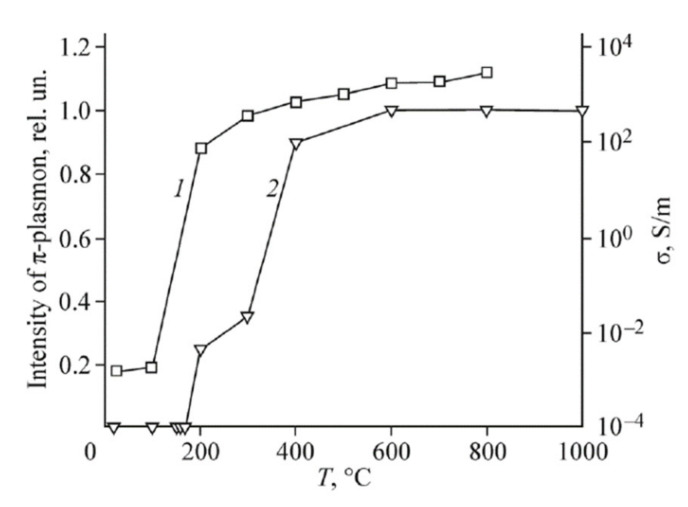
Comparison of dependences of the relative intensity of the π—plasmon peak (**1**) and the conductivity (**2**) of reduced GO sample on the annealing temperature [51].

**Figure 17 ijms-21-07634-f017:**
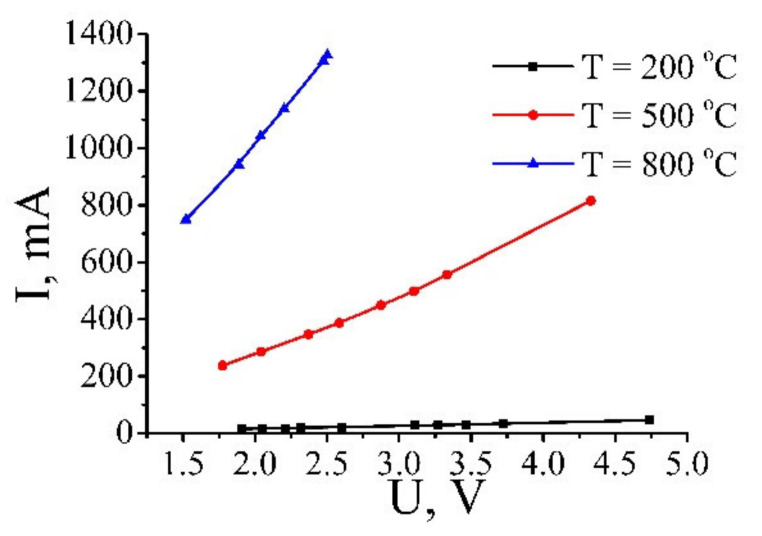
Current-voltage characteristics of GO samples annealed at various temperatures [19].

**Figure 18 ijms-21-07634-f018:**
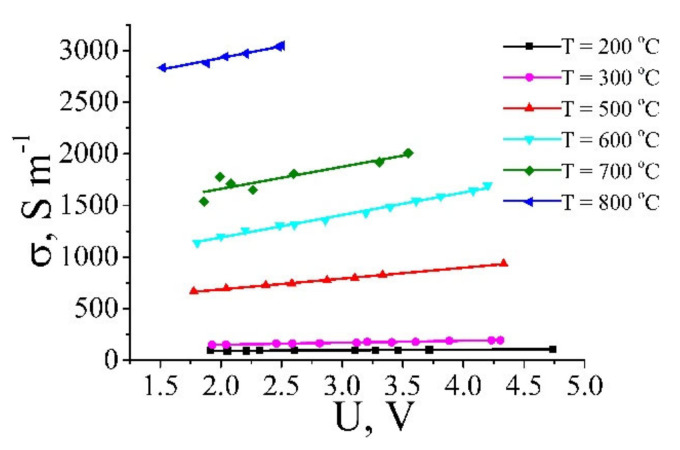
Dependences of the conductivity of GO samples annealed at different temperatures on the applied voltage. The data have been obtained in result of averaging the current-voltage characteristics for many samples [19].

**Figure 19 ijms-21-07634-f019:**
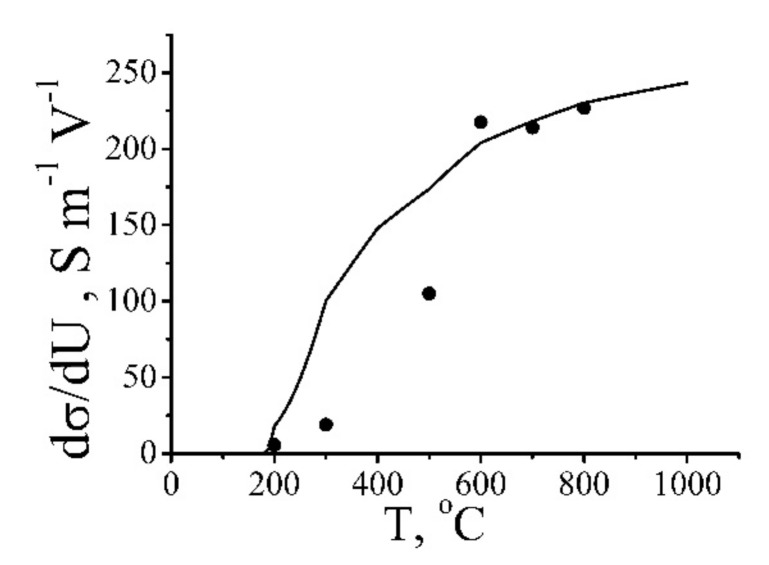
Points: Dependence of the empirical parameter *k*, determining the degree of non-linearity of conduction of reduced GO samples on the annealing temperature. Line: The result of calculation within the frame of the percolation model of conduction [20].

**Figure 20 ijms-21-07634-f020:**
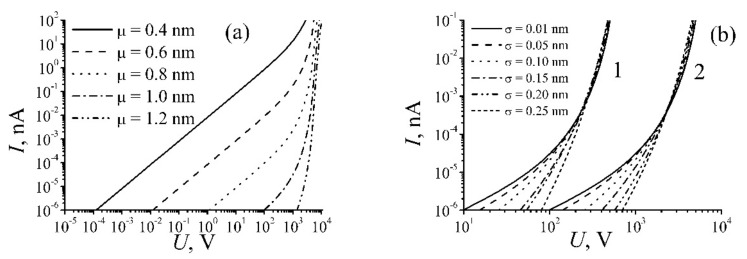
CVCs of percolation chains calculated for different values of parameters *σ*, *μ* and *n*: (**a**)—*σ* = 0.01 nm, the number of contacts *n* = 1000; (**b**)—*μ* = 1 nm, *n* = 100 (*1*) and 1000 (*2*) [21].

**Figure 21 ijms-21-07634-f021:**
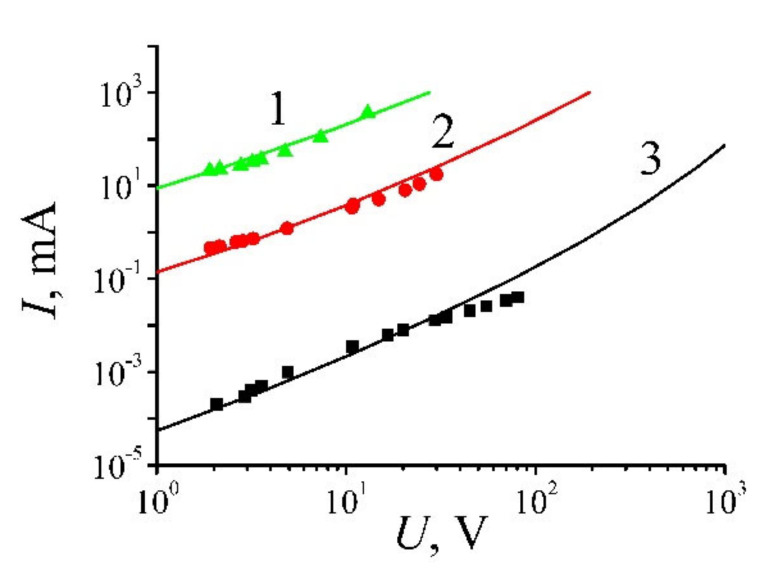
Current-voltage characteristics of OG samples annealed at different temperatures: Points—experiment; lines—calculations. (**1**) the annealing temperature *T* = 300 °C, µ = 0.422 nm; (**2**)—*T* = 200 °C, *µ* = 0.600 nm; (**3**)—*T* = 100 °C, *µ* = 0.925 nm; the standard deviation of the distribution (8) over the distances *σ* = 0.2 nm [21].

**Table 1 ijms-21-07634-t001:** Electronic characteristics of samples rGO annealed at different temperatures [46].

Annealing Temperature, °C	250	350	700	1000
Concentration and sign of current curriers, 10^16^ cm^−3^	1.99	−29.5	19.4	−60.1
Mobility of current carriers, cm^2^∙V^−1^∙s^−1^	2.56	5.48	62.8	188.0
Calculated conductivity, S/cm	0.05	0.12	1.75	9.44
Measured conductivity, S/cm	0.002	0.07	1.75	9.44

**Table 2 ijms-21-07634-t002:** Chemical composition of the initial sample and the samples annealed at different temperatures obtained from the analysis of XPS spectra, at % [19].

Annealing Temperature, °C	C,%	O,%	C/O	N,%	S,%	Si,%
25	74.7	23.0	3,25	1.3	0.5	0.4
150	73.6	25.1	2,93	0.7	0.5	-
200	82.0	15.2	5,39	1.6	0.5	0.7
600	90.6	8.1	11,2	0.5	-	0.7

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
