# Peer review of "Percolation Conduction of Carbon Nanocomposites"

_ijms, 2020, doi:10.3390/ijms21207634_

Round 1

Reviewer 1 Report

Review: Percolation conduction of carbon nanocomposites

This is a useful review, which presents theoretical approaches for modeling percolation behavior in composites with carbon fillers, such as CNTs and rGO. I believe the review needs addition of references to most recent works with graphene and related material for more balanced and up-to-date presentation. There are also a few technical points. I can recommend the review for acceptance after the authors made the following additions and addressed the following issues.

  • The differences between the aspect ration of CNTs and quasi-2D fillers like rGO should be emphasized more clearly and the effects of the aspect ratio on the percolation threshold described in more details.
  • There have been many papers on electrical and thermal percolation threshold with liquid-phase-exfoliated graphene and few-layer graphene which are better quality and more electrically conductive materials than rGO. The comparison with experimental works on percolation in graphene composites would make a lot of sense. The authors should cite the relevant papers (Z. Barani, et al., Adv. Funct. Mater., 1904008, 2019 – has both thermal and electrical percolation; “Thermal and electrical conductivity control in hybrid composites with graphene and boron nitride fillers,” Mater. Res. Express, vol. 6, no. 8, p. 085325, 2019; “Graphene epoxy-based composites as efficient electromagnetic absorbers in the extremely high-frequency band,” ACS Appl. Mater. Interfaces, vol. 12, no. 15, pp. 28635-28644, 2020; Adv. Electron. Mater., vol. 5, no. 1, p. 1800558, 2019) and compare the results when possible. Inclusion of experimental data on percolation in graphene composites would be useful. The inclusion of these recent works would also make the review more up to date.
  • Improve the figure reproduction when possible.

Author Response

    The differences between the aspect ration of CNTs and quasi-2D fillers like rGO should be emphasized more clearly and the effects of the aspect ratio on the percolation threshold described in more details.

The interconnection between the aspect ratio of fillers and the position of the percolation threshold has been noted in line 217 of the manuscript. For clarity sake, we inserted also a comment in lines 94-96.

    There have been many papers on electrical and thermal percolation threshold with liquid-phase-exfoliated graphene and few-layer graphene which are better quality and more electrically conductive materials than rGO. The comparison with experimental works on percolation in graphene composites would make a lot of sense. The authors should cite the relevant papers (Z. Barani, et al., Adv. Funct. Mater., 1904008, 2019 – has both thermal and electrical percolation; “Thermal and electrical conductivity control in hybrid composites with graphene and boron nitride fillers,” Mater. Res. Express, vol. 6, no. 8, p. 085325, 2019; “Graphene epoxy-based composites as efficient electromagnetic absorbers in the extremely high-frequency band,” ACS Appl. Mater. Interfaces, vol. 12, no. 15, pp. 28635-28644, 2020; Adv. Electron. Mater., vol. 5, no. 1, p. 1800558, 2019) and compare the results when possible. Inclusion of experimental data on percolation in graphene composites would be useful. The inclusion of these recent works would also make the review more up to date.

We supplemented the reference list with the references mentioned above accompanying them with short considerations.  

    Improve the figure reproduction when possible.

We tried to improve the figure reproduction.

Reviewer 2 Report

The proposed review paper appears as a collection of paragraphs without a general view in the field. It is clear that the authors do not represent expertise in the field to write a comprehensive review paper.

In fact the references lack of key works in the field and reports only week articles.

The number of figures is very high but again it is not clear the idea behind this work.

Author Response

The proposed review paper appears as a collection of paragraphs without a general view in the field. It is clear that the authors do not represent expertise in the field to write a comprehensive review paper.

In fact the references lack of key works in the field and reports only week articles.

The number of figures is very high but again it is not clear the idea behind this work.

The review does not contain specific remarks, however we have expanded the reference list including some recent publications (marked by red).

Reviewer 3 Report

Fig.19. Why the line corresponding to the result of calculation within the frame of the  percolation model of conduction is significantly different from the experimental points?

Please comment the relatively high percolation threshold for CNT based composites analyzed in the paper in comparison with published earlier (see e.g. Composites Science and Technology Volume 106, 16 January 2015, Pages 85–92  DOI information: 10.1016/j.compscitech.2014.11.004 2014).

Did you observe the re-organization of  GO particle and lowering the percolation threshold with the certain annealing temperature like it was observed in Composites Science and Technology, Volume 128, 18 May 2016, Pages 75–83, doi:10.1016/j.compscitech.2016.03.023 ? if not, please comment

One of the main aspects that influence the percolation processes is the way of nanocomposites processing (see e.g. Materials 2018, 11, 2256; doi:10.3390/ma11112256 and Materials (Basel). 2019 Jul 25;12(15). pii: E2369. doi: 10.3390/ma12152369). Please comment.

Did you check the aging behavior of your composites (see e.g. Composites Science and Technology 181:107712, DOI: 10.1016/j.compscitech.2019.107712)?

Author Response

Fig.19. Why the line corresponding to the result of calculation within the frame of the percolation model of conduction is significantly different from the experimental points?

We included the discussion of the comparison between the calculated and measured dependences of fig. 19 in lines 671-676. 

Please comment the relatively high percolation threshold for CNT based composites analyzed in the paper in comparison with published earlier (see e.g. Composites Science and Technology Volume 106, 16 January 2015, Pages 85–92  DOI information: 10.1016/j.compscitech.2014.11.004 2014).

We compare the position of the percolation threshold shown on fig. 4 with that measured in [25], One should note that the coincidence of this parameter for different composites is not necessary, because the position of the percolation threshold is determined by the aspect ratio of nanotubes which can be different for various experiments, depending on the filler’s aspect ratio.

Did you observe the re-organization of  GO particle and lowering the percolation threshold with the certain annealing temperature like it was observed in Composites Science and Technology, Volume 128, 18 May 2016, Pages 75–83, doi:10.1016/j.compscitech.2016.03.023 ? if not, please comment

We did not observe the percolation threshold lowering of partially reduced GO samples due to annealing, because in our experiments the percolation transition occurs as a result of the thermal reduction of GO, but not as a result of an increase of GO fraction. We mentioned the effect of the improvement of the quality of epoxy+GNP composites due to annealing in lines 362-365 (Ref. [34]).

One of the main aspects that influence the percolation processes is the way of nanocomposites processing (see e.g. Materials 2018, 11, 2256; doi:10.3390/ma11112256 and Materials (Basel). 2019 Jul 25;12(15). pii: E2369. doi: 10.3390/ma12152369). Please comment.

The interconnection between the shielding properties of GNP-filled composites and the way of their production found in [33] is commented in lines 356-361.

Did you check the aging behavior of your composites (see e.g. Composites Science and Technology 181:107712, DOI: 10.1016/j.compscitech.2019.107712)?

We have not noted aging phenomena in our composites, their electric characteristics remain to be stable for several months. We include the information on the aging obtained in [36] into the manuscript (lines 372-377).

All our corrections are marked with red.

Round 2

Reviewer 2 Report

Same as before.